# Near-Complete Response to Osimertinib for Advanced Non-Small-Cell Lung Cancer in a Pretreated Patient Bearing Rare Compound Exon 20 Mutation (S768I + V774M): A Case Report

**DOI:** 10.3390/ijms25147508

**Published:** 2024-07-09

**Authors:** Donato Michele Cosi, Cristina Fragale, Chiara Magri, Aldo Carnevale, Antonella Ciancetta, Massimo Guidoboni, Massimo Negrini, Giuseppe Bronte, Luana Calabrò

**Affiliations:** 1Department of Translational Medicine, University of Ferrara, 44121 Ferrara, Italy; donatomichele.cosi@edu.unife.it (D.M.C.); cristina.fragale@edu.unife.it (C.F.); chiara.magri@edu.unife.it (C.M.); aldo.carnevale@unife.it (A.C.); massimo.guidoboni@unife.it (M.G.); massimo.negrini@unife.it (M.N.); luana.calabro@unife.it (L.C.); 2Radiology Unit, University Hospital of Ferrara, 44124 Ferrara, Italy; 3Department of Chemical, Pharmaceutical and Agricultural Sciences, University of Ferrara, 44121 Ferrara, Italy; antonella.ciancetta@unife.it; 4Department of Oncology, University Hospital of Ferrara, 44124 Ferrara, Italy

**Keywords:** non-small-cell lung cancer, epidermal growth factor receptor, compound mutation, tyrosine kinase inhibitor

## Abstract

Third-generation tyrosine kinase inhibitors are the first-line gold standard in treating advanced non-small-cell lung cancer bearing common *EGFR* mutations, but data documenting clinical efficacy in uncommon mutations are currently limited. In this paper, we describe the case of a patient bearing uncommon compound *EGFR* mutations in exon 20, who experienced a near-complete response to third-line Osimertinib, with metabolic complete response of pulmonary, nodal and ostheolytic lesions. This radiological assessment corresponded to an ECOG PS improvement (from three to one) and a substantial clinical benefit for the patients. Out of two mutations, S768I was associated with poor response to third-generation TKI and V774M had unknown clinical significance, highlighting the complexity of the correct management of these kinds of mutations. We reviewed the literature to document the up-to-date preclinical and clinical data concerning third-generation tyrosine kinase inhibitors for the treatment of patients bearing uncommon *EGFR* mutations.

## 1. Introduction

The introduction of tyrosine kinase inhibitors (TKIs) has radically changed the systemic approach in treating oncogene-addicted non-small-cell lung cancer (NSCLC), achieving a selective inhibition of the mutated oncogene driver and its pathway. Specifically, third-generation epidermal growth factor receptor (EGFR) TKIs, as the first-line treatment of EGFR-mutated NSCLC, led to a consistent improvement in clinical outcomes. Nevertheless, this benefit is fully proven in patients bearing a common mutation, namely a deletion in exon 19 (ex19 del) or the point mutation in exon 21 (L858R). FLAURA trial, a pivotal phase III randomized clinical trial, showed that third-generation TKI Osimertinib was superior to second-generation TKIs in treating patients with advanced NSCLC bearing ex19 del or L858R mutation, resulting in a clinically meaningful advantage both in terms of progression-free survival (PFS) and overall survival (OS) [1]. As highlighted, this trial only enrolled patients bearing a common EGFR mutation, excluding patients with uncommon mutations, such as ex20ins, and other ex20, ex21 point mutations.

Even the more recent FLAURA-2, a phase III RCT evaluating the first-line combination of Osimertinib and platinum-based chemotherapy versus Osimertinib alone, excluded patients bearing uncommon or compound mutations [2].

As regards uncommon mutations, there are no solid data about the clinical efficacy of Osimertinib. Thus, at the state of the art, second-generation TKIs are the first-choice drug in treating EGFR uncommon mutations. Accordingly, the European Medicines Agency (EMA) approved Afatinib for all the EGFR-activating mutations, based on growing evidence of its activity on uncommon mutations [3,4].

Taking into account the toxicity and tolerability profile of second-generation TKIs, the use of a third-line TKI in patients bearing uncommon mutation would be an undoubtful advantage if its efficacy was pre-clinically and/or clinically proved [1].

Compelling evidence demonstrates that Osimertinib has a similar preclinical activity and clinical benefit in patients bearing some specific EGFR uncommon mutations compared to second-generation TKIs [5,6,7].

Given that, treating these patients in the optimal way is still an unmet need and poses a clinical challenge due to the complexity and rarity of the phenomenon and the subsequent lack of clinical data with sufficient levels of evidence.

Here, we present a case that occurred in our clinical practice of a patient, bearing uncommon ex20 compound mutations at baseline, who was treated with third-line Osimertinib after two lines of chemotherapy.

## 2. Case Description

An 81-year-old male patient referred to our ER on 11 August 2022 reporting subjective dyspnea.

His previous medical history included former smoker (45 p/y) status, essential thrombocythemia in hematological follow-up in therapy with hydroxyurea 500 mg bid and acetyl-salicylic acid 100 mg die, arterial hypertension under treatment with Ramipril, and radical prostatectomy 12 years before, followed by radiotherapy for adenocarcinoma in urological follow-up (Gleason Score 4 + 3, stable PSA suppression during previous follow-up).

He showed normal vital signs, specifically blood pressure 140/60 mmHg, oxygen saturation 95% (with no O_2_ supplementation), heart rate 91 bpm, and respiratory rate 21 per minute. The emergency doctor prescribed a chest X-ray that reported right basal pleural effusion. Because of his tachypnea, O_2_ therapy via nasal flow at 2 L/min was prescribed and the patient was admitted to the Pneumology ward for a better diagnostic framework. The total body CT scan confirmed massive right pleural effusion with multiple millimetric pulmonary nodules in the right lung and multiple bilateral mediastinal lymphadenopathies; brain and abdomen scans were oncologically negative, whereas bone examination reported multiple osteolytic areas along the spinal column (D8, D12, L4, L5 and S1). The following PET scan confirmed a contrast-enhanced area to the left pulmonary hilum and multiple nodal and skeletal areas of pathological uptake as metastasis, resulting in a cT4cN3cM1c Stage IVb tumor according to AJCC 8th (Figure 1). A right pleural drain was positioned and the pleural liquid underwent cytological examination that revealed the presence of TTF1+ tumor cells; the following bronchoscopy aimed to carry out histological characterization of a parietal pleural nodule confirmed the diagnosis of CK7+/TTF1+/WT1—adenocarcinoma, with immunohistochemistry (IHC) compatible with a pulmonary origin. Programmed death ligand 1 (PD-L1) on IHC revealed a tumor proportion score (TPS) of 2% and next-generation sequencing (NGS), performed on the tumor tissue, revealed two in cis mutations of the *EGFR* gene: S768I, with a variant allele frequency (VAF) of 43.2%; and V774M, with a VAF of 43.6%. The former mutation has a known pathogenetic meaning, whereas the latter has an unknown pathological meaning.

The patient, as highlighted, also had a previous prostate adenocarcinoma treated in 2010 with radical surgery and subsequent radiotherapy and underwent oncological follow-up for 5 years with no recurrence of disease evidence and stable PSA values (<0.05); for this reason, we did not take into account the double oncological diagnosis in evaluating the current oncological status.

Given the poor PS and the multiple comorbidities, after clinical discussion, the patients started first-line metronomic Vinorelbine on 28 September 2022. The treatment was well tolerated and led to a progressive clinical improvement. Because of hematological toxicity (g2 anemia and g1 thrombocytopenia), the hydroxyurea dose was reduced to 500 mg/die with a normalization of blood count. The first evaluation with total-body CT scan (16 December 2022) showed a skeletal and pulmonary PD.

The patient, given the improvement in PS and the good tolerance to the previous line, started second-line treatment with q21 AUC5 Carboplatin (with dose reduction due to hematological comorbidity) on 4 January 2023; even the second line was properly tolerated with the patient reporting only g1 asthenia. After six cycles of therapy, on 3 May 2023, a CT scan reported a further thoracic and skeletal progression (Figure 2).

Given the previous lines and the molecular characteristics of the disease, the patient started Osimertinib at 80 mg/die on 23 May 2023 along with iv q28 zoledronic acid. Treatment was well tolerated without relevant toxicity along the first two months of therapy. Furthermore, we observed a progressive improvement in Performance Status, with the patient reaching a stable PS 1 after three months of therapy, reporting gradually less dyspnea on exertion.

The first radiological evaluation occurred on 3 October 2023 and showed a clear reduction in pulmonary nodules as well as a skeletal disease progression on vertebral lesions (particularly S1); since the patient did not report any skeletal pain and we finally observed a pulmonary response, the treatment was continued beyond skeletal progression due to clinical benefit.

A second evaluation occurred on 1 December 2023, showing a metabolic complete response, both visceral and skeletal. The next radiological evaluation occurred on 15 March 2024, with a CT scan confirming a near-CR. The Appendix A demonstrate and describe radiologically the baseline burden of disease, the thoracic progressions to Vinorelbine and Carboplatin and the clear radiological response to Osimertinib (see Appendix A for further details).

The patient is currently still on treatment with Osimertinib at 80 mg/die with optimal tolerance and a current ECOG PS 1 due to comorbidity, reporting no pain together with stable clinical conditions.

## 3. Discussion and Literature Review

In order to better estimate the potential therapeutic value of the molecular analysis, our Molecular Tumor Board evaluated this case by using the Oncokb archive as a reference: “the EGFR exon 20 S768I mutation occurs in the EGFR tyrosine kinase domain. This mutation has been found in lung cancer [8,9,10,11,12,13,14,15]. Cell line experiments demonstrated that this mutation is activating [16] and transforming [17,18]. Some in vitro studies demonstrated that EGFR S768I confers resistance to EGFR TKIs with respect to other EGFR sensitizing mutations, such as L858R [16,17]. Preclinical models with the S768I mutation show resistance to third-generation EGFR inhibitors, moderate sensitivity to first-generation and Ex20ins-active EGFR inhibitors, and high sensitivity to second-generation EGFR inhibitors [19]. Preclinical models with the S768I and T790M mutations show resistance to first- and second-generation EGFR inhibitors, high sensitivity to third-generation EGFR inhibitors and moderate sensitivity to PKC and ALK inhibitors [19]. Reports of clinical sensitivity to TKIs in patients harboring the mutation have been varied [8,9,10,11,12,13,14,15]. Across eight clinical studies, thirteen patients with non-small-cell lung cancer (NSCLC) harboring the EGFR S768I mutation alone or in combination with erlotinib- and gefitinib-sensitive EGFR L858R, G719A or exon 19 deletion mutations had partial response (eight of thirteen), stable disease (two of thirteen) or progressive disease (three of thirteen), respectively, in response to treatment with gefitinib or erlotinib [8,9,10,11,12,13,14,15]. In a Phase II trial of osimertinib in Korean patients with NSCLC harboring uncommon EGFR mutations, three of eight patients with an S768I substitution had an objective response, with a median progression-free survival of 12.3 months [20]”. Moreover, in the EGFR exon, 20 insertion-mutated advanced NSCLC patients Amivantamab [21] and Mobocertinib [22] were tested in phase I-II clinical trials in patients with disease progression after a first-line platinum-based therapy.

In this case, the finding of compound mutations in exon 20 of the *EGFR* gene, with an already-known uncommon mutation (S768I) and a variant of unknown pathological meaning (V774M), together with advanced age and poor ECOG PS, led us to avoid Osimertinib as first-line treatment because we thought a more balanced risk-benefit could be achieved by giving vinorelbine [23]. We also avoided Afatinib, also effective for uncommon mutations, because of the high risk of diarrhea, which would have favored rapid clinical deterioration [24]. However, the evidence of progressive disease in the first CT scan, despite the initial improvement in symptoms, led us to treat the patient with carboplatin. Some evidence supports the use of platinum-based chemotherapy in elderly patients with good PS, but in this case, initial PS was poor and then improved; so, carboplatin only was our choice [25,26]. Finally, this option also had limited efficacy in terms of disease control, even though it was well tolerated. On the basis of EMA approval for Osimertinib in treating “activating EGFR mutations” (not approved by FDA for this label), we decided to start Osimertinib, instead of Afatinib, also taking into account the toxicity profile, given the patient’s initial poor PS, age and comorbidities.

Since the FLAURA trial showed a clear benefit in terms of both PFS and OS in advanced NSCLC harboring a EGFR common mutation [7], a remarkable effort has been made on studying Osimertinib efficacy even in uncommon EGFR mutations, including when these occur in combination.

As of May 2024, several case series and systematic reviews have well documented various uncommon EGFR mutations and their role in third-generation TKIs sensitivity, both via preclinical and clinical data.

In 2021, a post hoc subgroup analysis [27] pinpointed 21 out of 299 total patients from two Osimertinib phase II clinical trials (NCT02504346 and NCT03804580) bearing uncommon EGFR mutations. Among these patients, 38% (*n* = 8) had a G719X compound mutation with either S768I or L861Q as a partner mutation, resulting in an objective response rate (ORR) of 62.5% (95% CI: 24.5–91.5%) and a DCR of 100.0% (95% CI: 63.1–100.0%). The median DoR was 12.4 months, highlighting a more favorable outcome for patients with G719X compound mutations than for patients with other mutations.

In 2022, a systematic review [28] found 446 out of 46,679 patients presenting with multiple EGFR mutations, with as many as nine cases of triple mutations. This suggests it is a very rare event, mostly associated with the presence of T790M mutation. Preclinical data suggest that the presence of dual mutations may reduce the response to anti-EGFR [29,30].

The S768I mutation has a known pathological meaning: along with ex20 ins, this uncommon mutation has preclinically been linked with EGFR-TKI resistance [31,32]. Another systematic review analyzed over 6660 EGFR-mutated patients across 15 papers, reporting a prevalence of this mutation ranging from 0.5 to 2.5. Even if preclinical data may suggest TKI resistance, the literature reports’ contrasting data demonstrate a sporadic response to second- and third-generation TKI in some patients, presenting a S768I mutation both alone and as a compound [33].

A combined post hoc analysis reported an ORR of 100% among eight patients harboring the S768I mutation to Afatinib [24]. Another study reported an ORR of 25% to Icotinib for S768I mutation patients among 99 subjects harboring rare exon 20 mutations or insertion [34]. Similarly, a work with Korean patients reported a 38% ORR to Osimertinib (first line or pre-treated) to S768I single or associated with G719X in eight patients [20].

These data translated to a median PFS for the previously described patients of 14.7mo (95% CI 2.6-NE), 2.0mo (95% CI NR), 12.3mo (95% CI 0–28.8), respectively, for the three aforementioned studies (Table 1).

The V774M mutation has an unknown clinical meaning; across the literature, a single case report has documented Afatinib resistance in V774M-H773L compound mutations [35].

Given that these two mutations were never found together in a patient, and we observed a good response to osimertinib, we decided to perform a molecular modeling analysis of the EGFR protein bearing these modifications and its interaction with the inhibitor. Three-dimensional models of both wild-type and S768I/V774M mutant EGFR tyrosine kinase domain (limited to the P694-G1021 portion) were built using the AlphaFold 3 web-server [36] and by selecting ADP among available co-folding ligands. Osimertinib coordinates were imported in the models by superimposing the models to the X-ray complex with EGFR (L858R/T790M/C797S) available in the Protein Data Bank (PDB ID: 6LUD) [37] by ensuring the key inhibitor–protein interactions as observed in the experimental structure were preserved and no significant clashes were introduced. The position of S678I and V774M mutations (purple spheres in Figure 3 left) with respect to Osimertinib binding site (blue surface in Figure 3, ligand in magenta sticks) and their putative effect with respect to the wild-type model in the overall kinase domain architecture were inspected. As depicted in Figure 3 (left panel), both mutations insist on the loop following the α-C helix (green cartoons in Figure 3), which is key for EGFR activation. In particular, the S678I mutation (upper right panel in Figure 3) is expected to unhinge the α-C helix position by disrupting a network of H-bonds connecting S768 (yellow sticks in Figure 3) to Asn700 and Arg831 in the wild-type model, whilst the V774M mutation (lower right panel in Figure 3) is predicted to create steric bulk with nearby residues in the α-E helix. We speculate that these structural changes would push the α-C helix away from its position and stabilize a α-C helix-out conformation, therefore impacting enzyme activity whilst still enabling osimertinib to be accommodated in the active site and exert its function. (Figure 3).

## 4. Conclusions

Increasing evidence from retrospective data and case reports supports the efficacy of Osimertinib in treating uncommon EGFR mutations, including those occurring alongside common, well-known mutations. However, scientific evidence from prospective trials is still lacking, likely due to the rarity of these mutations. To the best of our knowledge, this is the first case report describing the clinical efficacy of Osimertinib in a patient with the S768I and V774M EGFR compound mutation, who was also pretreated with chemotherapy.

## Figures and Tables

**Figure 1 ijms-25-07508-f001:**
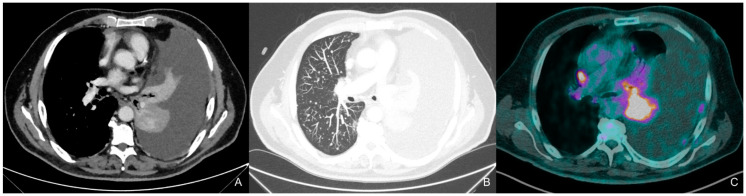
CT 18/08/22, PET-CT 22/08/22, the contrast-enhanced chest computed tomography (CT) scan, obtained in the venous phase and visualized using the mediastinal window (**A**), reveals a left hilar pulmonary mass causing complete left lung atelectasis, accompanied by ipsilateral pleural effusion with irregular parietal pleural thickening, consistent with pleural carcinomatosis. Maximum intensity projection (MIP) reconstruction in the lung window (**B**) demonstrates multiple nodules and micronodules randomly distributed in the right lung. The positron emission tomography (PET)-CT scan (**C**) shows avid uptake of the radiotracer in the lung mass and in a right hilar lymph node (10R station).

**Figure 2 ijms-25-07508-f002:**
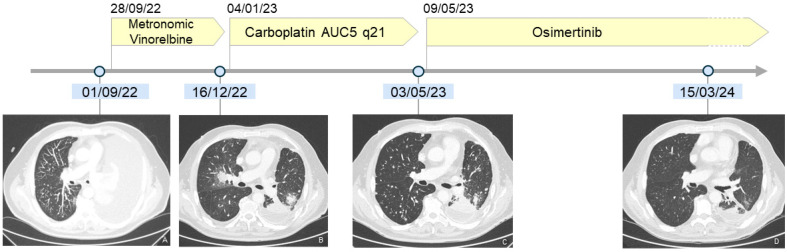
Timeline summarizing patient’s history and documenting minimal response to chemotherapy and radiological near-CR to Osimertinib. (**A**): see Figure 1; (**B**): (CT 16/12/22) the chest CT scan, obtained after left pleurodesis, shows at the hilum level the pulmonary progression to Vinorelbine; (**C**): (CT 03/05/23): the chest CT scan demonstrates a further increase in number and size of the lung nodules after Carboplatin administration; (**D**): (CT 15/03/24): the chest CT scan shows the resolution of the lung nodules and a significant decrease in the previously evident lung consolidations at the hilum. See Appendix A for further details on the disease course during the various lines of therapy.

**Figure 3 ijms-25-07508-f003:**
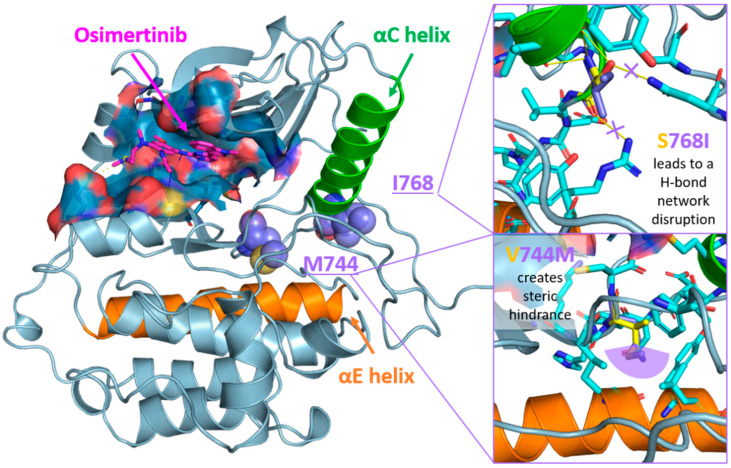
**Left panel**: position of I768 and M744 mutations (purple spheres) with respect to Osimertinib binding site (blue surface, ligand in magenta sticks) in the EGFR kinase domain structure. Inhibitor–enzyme interactions are color-coded as follows: hydrophobic, dashed grey lines; hydrogen bond, solid blue lines; salt bridge, yellow dashed lines. The α-C and α-E helixes are highlighted as green and orange cartoons, respectively. **Right panels**: detail of the expected effect of S768I (**upper panel**) and V744M (**lower panel**) mutations. Residues in the WT and S768I/V744M EGFR structures are rendered as yellow and purple sticks, respectively. Position of residues surrounding the mutation sites in the WT model are represented as cyan sticks. H-bond and steric hindrance are visualized as yellow solid lines and a purple semi-circle, respectively. Three-dimensional models were generated with AlphaFold3 web-server [35], ligand–active site interactions were inspected with PLIP [38] and the images were rendered with PyMOL open source v2.5.0.

**Table 1 ijms-25-07508-t001:** Clinical outcomes of uncommon EGFR mutations, treated with TKI.

Article Reference	Article Type	*N* of Cases	EGFR Mutation	TKI	Line of Therapy	ORR (%)	PFS (months)
J. C.-H. Yang et al. [24]	Post hoc analysis of clinical trials (LUX-Lung 2, 3 and 6)	75	S768I +/− G719X or L858A (*n* = 8)	Afatinib	Any	100%	14.7 (95% CI 2.6-NE)
C. Xu et al. [34]	Retrospective study	99	S768I (*n* = NR)	Icotinib	Not reported	25%	2.0 (95% CI NR)
J. H. Cho et al. [20]	Prospective study	37	S768I +/− G719X (*n* = 8)	Osimertinib	Any	38%	12.3 (95% CI 0–28.8)
I. J. Z. Eide et al. [27]	Post hoc subgroup analysis	21	G719X compound mutation with either S768I or L861Q (*n* = 8)	Osimertinib	Any	62.5%	13.7 (95% CI NR)

## Data Availability

More detailed information of the patient supporting the conclusions of this article are available on request from the corresponding author. The data are not publicly available due to patient’s privacy.

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
