# Peer review of "Near-Complete Response to Osimertinib for Advanced Non-Small-Cell Lung Cancer in a Pretreated Patient Bearing Rare Compound Exon 20 Mutation (S768I + V774M): A Case Report"

_ijms, 2024, doi:10.3390/ijms25147508_

Round 1

Reviewer 1 Report

Comments and Suggestions for Authors

This is a case report manuscript.

Patient Information:

  • Age: 81 years old
  • Medical History:
    • Former smoker (45 pack-years)
    • Essential thrombocythemia (hydroxyurea 500mg bid, acetylsalicylic acid 100mg daily)
    • Hypertension (treated with Ramipril)
    • Radical prostatectomy (12 years ago), followed by radiotherapy for adenocarcinoma

Initial Presentation:

  • Date: August 11, 2022
  • Symptoms: Subjective dyspnea
  • Vital Signs:
    • BP: 140/60 mmHg
    • Oxygen saturation: 95% (no O2 supplementation)
    • Heart rate: 91 bpm
    • Respiratory rate: 21 per minute
  • Findings: Right basal pleural effusion (chest X-ray)

Admission and Diagnosis:

  • Admitted to: Pneumology ward
  • CT Scan:
    • Right pleural effusion
    • Multiple pulmonary nodules (right lung)
    • Bilateral mediastinal lymphadenopathies
    • Negative brain and abdomen scans
    • Multiple osteolytic bone areas
  • PET Scan: Metastasis confirmed, cT4N3cM1c Stage IVb tumor (AJCC 8th)

Immunohistochemistry revealed a PD-L1 tumor proportion score of 2%, and next-generation sequencing identified two EGFR mutations: S768I and V774M. The Molecular Tumor Board utilized the OncoKB archive to interpret these mutations, noting that S768I is activating and confers resistance to certain EGFR inhibitors while showing varied clinical responses. This case highlights the complexity of managing advanced-stage lung cancer with specific genetic mutations and underscores the importance of precision medicine in guiding treatment strategies.

1.      Some discussion was taken in the case description. Please move them to the third part of this case report.

2.      Please consider provide CARE checklist, using it can significantly improve the quality and clarity of your case report, making it more valued by the medical community.

3.      Please add a table as a summary of previous reported rare cases. Including a table can greatly enhance the clarity and utility of your literature review by providing a quick reference for the key information. You can summarize key findings from previous case reports and studies. Most importantly, please compare the current case with existing literature and then highlight similarities and differences between the current case and previously reported cases.

4.      Because this patient has had prostate cancer, please also discuss about double cancer in this case.

5.      Did the old age interfere with your clinical decision making?

Author Response

Reviewer 1

This is a case report manuscript.

Patient Information:

Age: 81 years old

Medical History:

Former smoker (45 pack-years)

Essential thrombocythemia (hydroxyurea 500mg bid, acetylsalicylic acid 100mg daily)

Hypertension (treated with Ramipril)

Radical prostatectomy (12 years ago), followed by radiotherapy for adenocarcinoma

Initial Presentation:

Date: August 11, 2022

Symptoms: Subjective dyspnea

Vital Signs:

BP: 140/60 mmHg

Oxygen saturation: 95% (no O2 supplementation)

Heart rate: 91 bpm

Respiratory rate: 21 per minute

Findings: Right basal pleural effusion (chest X-ray)

Admission and Diagnosis:

Admitted to: Pneumology ward

CT Scan:

Right pleural effusion

Multiple pulmonary nodules (right lung)

Bilateral mediastinal lymphadenopathies

Negative brain and abdomen scans

Multiple osteolytic bone areas

PET Scan: Metastasis confirmed, cT4N3cM1c Stage IVb tumor (AJCC 8th)

Immunohistochemistry revealed a PD-L1 tumor proportion score of 2%, and next-generation sequencing identified two EGFR mutations: S768I and V774M. The Molecular Tumor Board utilized the OncoKB archive to interpret these mutations, noting that S768I is activating and confers resistance to certain EGFR inhibitors while showing varied clinical responses. This case highlights the complexity of managing advanced-stage lung cancer with specific genetic mutations and underscores the importance of precision medicine in guiding treatment strategies.

  1. Some discussion was taken in the case description. Please move them to the third part of this case report.

REPLY: Molecular tumor board discussion moved to the third part.

  1. Please consider provide CARE checklist, using it can significantly improve the quality and clarity of your case report, making it more valued by the medical community.

REPLY: CARE checklist provided

  1. Please add a table as a summary of previous reported rare cases. Including a table can greatly enhance the clarity and utility of your literature review by providing a quick reference for the key information. You can summarize key findings from previous case reports and studies. Most importantly, please compare the current case with existing literature and then highlight similarities and differences between the current case and previously reported cases.

REPLY: Table added in discussion

  1. Because this patient has had prostate cancer, please also discuss about double cancer in this case.

REPLY: We did take into account the double cancer, but given that radical surgery + RT for prostate cancer occurred more than 10 years before the pulmonary cancer onset we did not change our evaluation of the patient, due to his multiple, active comorbidities. We added a small discussion in the case presentation.

  1. Did the old age interfere with your clinical decision making?

REPLY: the old age and poor performance status at diagnosis did interfere with our clinical decision making in offering first-line metronomic Vinorelbine, also considering the hematological comorbidity. We still decided to offer second-line AUC 5 q21 Carboplatin, as the patient’s PS improved during the treatment, still in monotherapy, given the old age and comorbidities. We did not take into account the old age when we offered third-line Osimertinib, based on literature data indicating a tolerable safety profile in fragile patients (e.g. https://pubmed.ncbi.nlm.nih.gov/30651400/): we did not decide to propose upfront Osimertinib exclusively due to poor efficacy data across literature on the two specific EGFR mutations found in this case.

Reviewer 2 Report

Comments and Suggestions for Authors

Dear Editor and Authors,

I would like to express my gratitude for the opportunity to review this manuscript 

This is an important case report with significant clinical relevance and well-described rationale. I have only a few minor suggestions for improvement:

Abstract is too short. The "near complete response" can be more specified in clinical and radiological terms. A brief statement on the outcome and implications for clinical practice could also be included.

Include in the introduction a brief mention of the rarity and clinical challenges associated with compound EGFR mutations, as this sets the context for the case report. 

The objectives in introduction are not clear and should include the literature review to agree with the title. Or change the title removing “a literature review”. Additionally, the “2. Case description” should be “2. Case description and literature review”

Add the sex of the patient in the first statement of case description to be clear and not implicit.

The discussion could benefit from a more focused comparison with other reported cases of compound EGFR mutations treated with Osimertinib, for example: PMID: 31564718.

Author Response

Reviewer 2

Dear Editor and Authors,

I would like to express my gratitude for the opportunity to review this manuscript

This is an important case report with significant clinical relevance and well-described rationale. I have only a few minor suggestions for improvement:

Abstract is too short. The "near complete response" can be more specified in clinical and radiological terms. A brief statement on the outcome and implications for clinical practice could also be included.

REPLY: abstract modified as requested

Include in the introduction a brief mention of the rarity and clinical challenges associated with compound EGFR mutations, as this sets the context for the case report.

REPLY: implemented as requested

The objectives in introduction are not clear and should include the literature review to agree with the title. Or change the title removing “a literature review”. Additionally, the “2. Case description” should be “2. Case description and literature review”

REPLY: we changed the title of the 3rd section (discussion and literature review), where the table added as suggested by Reviewer 1 synthesizes the papers considered in the review

Add the sex of the patient in the first statement of case description to be clear and not implicit.

REPLY: We added the sex of the patient (male) in case description

The discussion could benefit from a more focused comparison with other reported cases of compound EGFR mutations treated with Osimertinib, for example: PMID: 31564718.

REPLY: We added a table to compare the findings from studies with compound mutations. We cited the article PMID: 31564718 as regards resistance mechanisms.
